# Construction of an Ophthalmological Calendar for the Therapeutic Follow-Up of Glaucoma in the Elderly

**DOI:** 10.3390/ijerph20021237

**Published:** 2023-01-10

**Authors:** Margarida da Silva Neves de Abreu, Maria Clara Palitot Galdino, Suellen Duarte de Oliveira Matos, Carla Christina de Lima Pereira Bezerra Cavalcanti, Débora Raquel Soares Guedes Trigueiro

**Affiliations:** 1Escola Superior de Enfermagem do Porto, 4200-072 Porto, Portugal; 2Programa de Pós Graduação de Mestrado em Saúde da Família da, Faculdades de Enfermagem e Medicina Nova Esperança, João Pessoa 58067-698, PB, Brazil; 3Centro de Ciências Médicas, Universidade Federal da Paraíba, João Pessoa 58051-900, PB, Brazil

**Keywords:** health education, glaucoma, health service for the elderly

## Abstract

Health teams in primary care play a key role in the eye health of users as they understand that early identification of any visual change can result in satisfactory outcomes and better prognoses, preventing damage that is often irreversible to health**.** Building an ophthalmological calendar for the therapeutic follow-up of glaucoma in the elderly, this is a methodological study, as the process of constructing the calendar’s content followed the Raymundo theoretical framework. The calendar was built in the following steps: bibliographic survey, content development, transformation of the language of scientific information into easy-to-understand expressions, creation and production of illustrations of the first draft, evaluation of the first draft made by the examining board, diagramming and presentation of the product. The construction of the calendar covers a specific theme for the elderly with glaucoma, which emphasizes the need to invest more in the inclusion of new technologies that will provide greater effectiveness and adherence of the user and the health team for the management of comprehensive care. The implementation of the produced calendar will allow for a better understanding and bond between the team professionals and the user and, consequently, a better monitoring of the therapeutic process of the patient involved.

## 1. Introduction

Glaucoma is an optic neuropathy of multifactorial, asymptomatic, chronic, hereditary, bilateral and asymmetrical etiology, whose diagnosis is made through the evaluation of parameters seen in routine eye exams; progressive loss of the visual field is confirmed with a complementary exam [1].

Overall, the current mainstay of treatment for all glaucoma patients is the preservation of visual function while maintaining the best possible quality of life. Currently, the aim is to modify the progression of the glaucomatous lesion through the reduction of IOP (intraocular pressure) to achieve minimal additional damage to the optic nerve, since it is a chronic disease [2].

Health teams in primary care plays a key role in the eye health of users as they understand that early identification of any visual change can result in satisfactory outcomes and better prognoses, preventing damage that is often irreversible to health [3].

In chronic diseases, daily treatment is inevitable, and it is necessary to give the best possible information to the patient about the treatment and its risks. Improper technique, medication loss, overdose with systemic absorption, adverse effects, predisposition to infection by contamination of the eyedrops tip, corneal abrasion and ulcers are frequent, requiring a link with the elderly to investigate the problem and implement it more strategically [4].

The use of eye drops is a common practice in patients with glaucoma. When observing the instillation technique of this medication, it is often perceived that it is misused, risking infection by contamination, corneal abrasion and ulcers, particularly in the elderly [5].

For these reasons, it is important for the elderly person or caregiver to have a record of notes in large printed material, to allow the patient to see how many times they have used the eye drops and differentiate the types of medication. Therefore, the study aimed to build an ophthalmological calendar for the therapeutic follow-up of glaucoma in the elderly.

## 2. Materials and Methods

To substantiate this technique, the research was carried out in two stages. In the first, an exploratory, descriptive and quantitative research was carried out. For this first stage, the study was developed at Family Health Units, which is linked to Primary Health Care in District III of the city of João Pessoa-PB, Brazil. The sample consisted of 61 elderly people who assisted in the study setting. As the research instrument, a questionnaire was created with questions based on the objectives of the study. Descriptive statistics were used for data analysis (absolute and relative frequency). In the main findings of this stage regarding the treatment for glaucoma, 31 (50.8%) individuals used eye drops for more than 2 years, 46 (75.4%) did not know the name of the eye drops, 41 (67.2%) already dripped the eyedrops more than once in the same eye because of lack of perception and 42 (70.5%) used eyedrops more than once per day. 

In the second stage, for the construction of the calendar and purposes of this article, a methodological study was chosen, which was developed from February to April 2021. The calendar content construction process followed the theoretical framework concerning the first version (“item generation”, “error collection” and “analysis of redundancy added to the composition and validation of content”), and was determined by Raymundo [6].

In addition to these steps, a construction process was necessary for the refinement of the technology developed in seven steps: 1. bibliographic survey; 2. content development; 3. transformation of the language of scientific information into easy-to-understand expressions; 4. creation and production of the first draft illustrations; 5. evaluation of the first draft made by the examining board; 6. technological diagramming; and 7. technological product presentation.

Step 1—Bibliographic survey: As a contribution to support the content covered by the educational technology (calendar), a survey was carried out of the updated publications of the Ministry of Health until 2019 and the Brazilian Society of Ophthalmology (2019).

Step 2—Content development: In this phase of the calendar construction, there was a need to carry out an active search in databases, such as the Jaeger table, publications referring to the contrast threshold for angular frequencies, as well as selection of studies that covered the content proposed in this research. After analyzing the bibliographic productions, three publications were selected that helped in the theoretical-scientific foundation of the calendar and subsidized the adequacy of texts and images. For elaboration, the information was organized step by step for instillation: adequate hygiene, checking the medication referring to the ophthalmologist’s prescription, dosage, instructions on how to apply it correctly so that there is effective absorption of the eye drops, adequate closure of the medication, filling in the calendar at the time of application, necessary care to assist the Basic Health Unit team (UBS—in Portuguese) in verifying the use, as well as analysis of the most common errors in the application of eye drops during treatment, based on the perspective of promoting the autonomy of the elderly, engagement and strengthening of the service.

Step 3—Transformation of the language of scientific information into expressions that are easy to understand: In order to transform the language of scientific information into expressions that are easy to understand for the target audience, it was necessary to readjust the more elaborate language to expressions that are easy to understand, in order to facilitate the assimilation and completion of the calendar by the elderly person and their caregiver.

Step 4—Creation and production of the illustrations of the first draft: The first draft was drawn with the letters in large size for the patient. For this purpose, the Jaeger table was used to investigate the near vision, with the letters of the size corresponding to J4 to J5 (Figure 1), in addition to the black font with a white background or coloring. For the elaboration of the therapeutic calendar, the Snellen scale of vision 20/60 was used for visual acuity for distance, in large letters and with high contrast (black/white). The calendar contains space to write down doubts or complaints so that they can be clarified at the next follow-up appointment at the specialized service, as well as an area to schedule medication use at times according to medical prescription. It also has colored markers containing primary and distinct colors so that they are associated with the color of the eye drops (each set of eye drops will be marked according to the active principle of the prescribed medication). There are also captions so that the family and community health team can verify correct use, common side effects of the medication and the complaints reported by the elderly. The color scheme adopted in the calendar will facilitate the understanding of the information for literate and non-literate elderlies.

Step 5—Evaluation of the first draft made by the examining board: The examining board, formed by two ophthalmologists and two nurses, evaluated the graphic designs, layout and information covered in the calendar in order to organize all material specifications. The necessary changes were made according to the suggestions of the examiners.

Step 6—Diagramming: The elaborate calendar was sent to the graphic designers, who received guidance so that the illustrations were attractive and easy to understand. These diagrammers were responsible for the elaboration of graphic designs, calendar layout, involving formatting, configuration and layout of the pages. The production of the material was developed by a DELL brand laptop, using the software Adobe Illustrator 2020, as well as by a digital table of the brand Wacom. The typography (font) used was the Museo Family, as it provides good reading and presents an extensive variety of types. The colors used were red, yellow and green, with well-differentiated shades to “give more life” to the material and bring a more youthful appearance. The aesthetic composition was thought to be an attractive material, which did not only have a medical function, but functioned as a visual element, which adds value and beauty to the patient’s home and “shines in the eyes” of their visits. The illustrations representing the elderly in the calendar sought to maintain an aesthetic following the graphic design, which was focused on bringing a youthful air to the instrument and, consequently, beauty to all material, promoting a visual association of the user’s image with the proposed calendar. One of the predominant graphic elements is the representation of an eye through a minimalist style. This choice is justified because of the intention to create 38 units with the entire project, correlating the proposal (calendar) with its function in ophthalmology.

Step 7—Presentation of the technological product: The material consists of a folder for the elderly to carry the calendar during consultations. On the outside of the folder, there is a space for patient identification data (name, address and telephone), as well as the title of the proposed technology: “My Glaucoma Calendar”. Inside the kit, there are two booklets: a larger one, containing instructions for the healthcare professional who is going to handle the instrument. The Calendar (Figure 2) has a place for marking with stickers or circular labels to make it easier to fill in the information. In the posterior portion, there are three columns: in the first there, is a place for the user to take notes; in the second, there are instructions for the application of the medication and the use of the instrument; and, finally, in the third and last column, there are instructions for the health professional on taking notes.

Finally, “My Glaucoma Calendar” technology obtained the registration of titles and documents registered in Book B 6395 under No. 797416 at the Toscano de Brito registry office—Notary and registry service. João Pessoa – PB. 04/06/2021 15:16:35. DIGITAL SEAL: ALG84145- WNWB.

## 3. Results

### Calendar

The calendar, aimed at the elderly and caregivers to overcome their difficulties, through a soft–hard technology proposal, provides large, colourful, educational and easy-to-use printed material for filling in the information and notes of doubts about the use of eye drops. It is important to mention that the use of the therapeutic calendar aims to help health professionals in monitoring effective treatment, as well as guiding patients.

O kit vem direcionado para o profissional de saúde da atenção primária, contendo informações e instruções que poderão ser de fácil seguimento na esfera de cuidado da ESF. É também direcionado ao oftalmologista, que certamente terá um melhor controle da resposta terapêutica, podendo ser útil para desenvolver um melhor vínculo com seu paciente, dinamizando mais as consultas de acompanhamento, uma vez que a ferramenta facilita a integração do idoso em seu próprio tratamento. The Calendar Kit consists of: a calendar with information on the front and back, N of stickers (commercial circular envelope label, in the dimensions of 1 × 1 cm), 200 *g* coated paper folder to completely wrap the material for transport with the proposed layout (envelope), being that the material is of greater durability. There will also be an explanatory insert for the professional and another one for the user/or companion with a presentation of the kit for both.

The kit will be delivered by the specialized secondary action professional, who will mark the eye drops with the color corresponding to the mechanism of action present in the medication, following the instructions in the intended booklet. The professional must also add the same sticker marking to the prescription, and later, he must deliver the booklets with the same color for the patient to use in the daily marking.

For the printing of the material, the following specifications are suggested: for the envelope, a 180 *g* coated paper, colored, double-sided, 320 mm wide × 285 mm high, should be recommended. The calendar can be printed on an A3 sheet (proposed size) or 04 A4 sheets of plain paper. For greater durability of the instrument and removal of the stickers without harming the technological tool, it is necessary that the printing of the material is carried out on 180 *g* or 200 *g* coated paper. Printing on plain paper is allowed, provided that it is laminated (which, however, would make it impossible to transport it in the envelope); in any material chosen, the calendar must be 420 mm wide × 297 mm high with a colored front and back. For printing the instruction to Family Health Team (FHT), it is recommended that it be printed on both sides, in color, on an A4 sheet, 210 mm wide × 297 mm high. As for the instructions for users, the print will be single-sided, in color, size 150 mm wide × 210 mm high and equivalent to half the dimensions of an A4 sheet. 

## 4. Discussion

This calendar is intended for everyone involved in the care of the elderly person with glaucoma, that is, the patient himself. It is an instrument that proposes to maintain a communication channel and promote better management of this user in the care network proposed by the National Health System (SUS—in Portuguese), being a means of connecting the primary care health professional with the ophthalmologist and, with the patient, in order to promote a unified approach to the treatment of the elderly.

The kit is aimed at the primary care health professional, containing information and instructions that can be easily followed in the sphere of FHT care. It is also directed to the ophthalmologist, who will certainly have better control of the therapeutic response, which can be useful to develop a better bond with the patient, making follow-up consultations more dynamic, since the tool facilitates the integration of the elderly in their own treatment. On the other hand, it is also an instrument that belongs to the elderly, that is, the patient will have a very dynamic calendar, with a proposal to be happy and that can be exposed in the home environment involving their family and friends.

Brazil is one of the countries with the most publications on the inclusion of technologies that promotes education for the elderly and the community. The justification for this is due to the prioritization of actions aimed at the health of the elderly based on the National Agenda of Priorities in Health Research, which proposes an active role for the elderly in order to optimize health strategies [7]. 

The integrative review study developed by Sá [7] aimed to identify the use of soft, soft–hard and hard technologies in the care of the elderly and to analyze how the multiple forms of technology contribute to the care of the elderly in health services. The study concludes that the development of technologies aimed at the elderly public is considered as an action that promotes advances in science in the field of health, and that the construction of technology should encompass a multidimensional knowledge, with easy replicability and low cost, promoting, in this way, adherence to health services.

In order to adhere to the FHT calendar, it is important to emphasize that the technological product has many self-explanatory images and captions aimed at primary care health professionals, in order to facilitate the bond between the elderly and health professionals and to strengthen health care. Therefore, actions that consider the integrality of the patient and shows an active role for the elderly should optimize strategies for health promotion and dialogic integration between the health professional and the elderly person and his/her family [8].

Another integrative review study, whose objective was to identify scientific evidence about educational technologies in promoting the health of the elderly, showed the need to develop strategies aimed at the integration of innovative educational technologies to promote health care. Soft and soft–hard technologies are more accepted among the elderly public and health professionals [9].

In relation to the health team, this procedure will contribute to the guidelines during care, in addition to providing a unified language. It is noteworthy that this vivid and strong color scheme was used so that the patient could more easily identify the information contained therein. A caption was placed to correlate the eye drops with the way it is being used by the patient, with its possible side effects and some more observations that can be described in a scientific way so that health professionals can follow up.

The construction of the calendar covers a specific theme aimed at the elderly public with glaucoma, which emphasizes the need to invest more in the inclusion of new technologies that will provide greater effectiveness and adherence of the user and the health team for the management of comprehensive care. In this sense, producing more soft–hard technologies in the health field will strengthen more assertive practices.

Since in the context of glaucoma in the elderly, no technology with the structure of a therapeutic schedule was found available in the literature, our proposal presents a technique and identifies the benefits of adherence and its use. The benefits can be summarized as: better monitoring of users by the family health team in relation to the use of eye drops and reorganization of the care process for these users, ensuring a better quality of life. In this sense, the construction and dissemination of health technologies has proved to be of great importance, which is why it is intended, later, to submit the technological product for evaluation by experts in the area for refinement.

## 5. Conclusions

It is noteworthy that elderly people with glaucoma have disadvantages when compared with other citizens, not only because of the condition of the disease, but also because of the lack of information about their potential. Thus, health education for the elderly and their families, individually or in groups, is presented as a strategy that enables participation and involvement in therapy.

In this way, this study aimed to build a calendar, with the aim of helping the elderly and/or their caregivers in the monitoring and verification of the treatment performed by the family health unit team and thus help in making more assertive decisions about health. of the elderly. In this sense, it is necessary to implement the calendar produced by this study, as it will allow for a better understanding and bond between the team professionals and the user and, consequently, a better monitoring of the therapeutic process of the patient involved.

Therefore, it is recommended that further investigations on the subject make use of this technological tool in order to minimize damage to users and reorganize work processes with a focus on health care. Such measures are urgent to consolidate health practices focused on the user**’**s well-being, as recommended by the family health strategy model linked to the Unified Health System.

The calendar is an instrument for the elderly to use based on their autonomy during the treatment of glaucoma—technical information such as intraocular pressure and perimetry are dispensed from the instrument. Such parameters are important for the ophthalmologist to assess the evolution of the health condition. These results will be saved in the patient**’**s medical record and delivered to him regularly. The calendar aims to help the ophthalmologist to assess adherence to treatment, an important factor that explains the prognosis.

As for the limitation of this research, we highlight the fact that there was no validation by specialists, a very important factor in the construction of a technology focused on health, since this validation will allow the refinement of the constructed material and, thus, provide far-fetched adaptations that more effectively meet the needs of this population.

## Figures and Tables

**Figure 1 ijerph-20-01237-f001:**
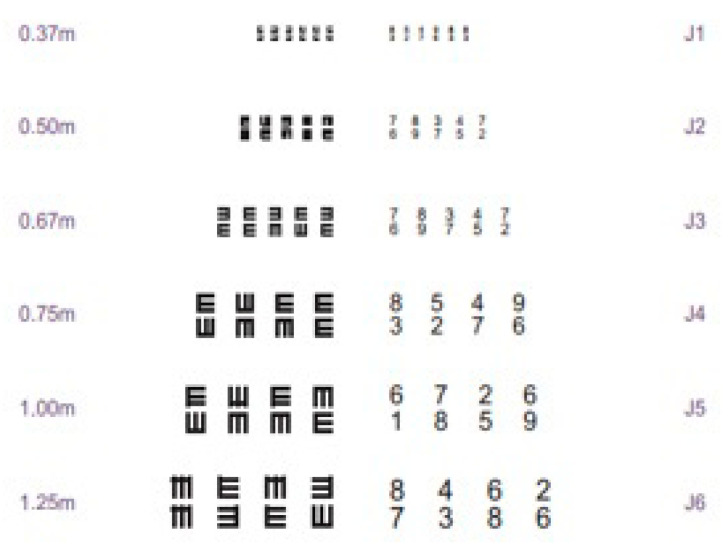
Jaeger’s Table for Measurement of Near Vision. (Source: Costa; Santos, 2018).

**Figure 2 ijerph-20-01237-f002:**
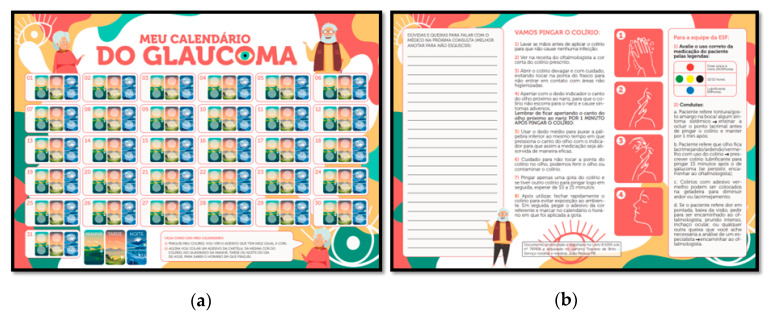
Final proposal for the construction of the technological product “My Glaucoma Calendar”. João Pessoa, Paraíba, Brazil, 2020: (**a**) front; (**b**) back.

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
