# Peer review of "Construction of an Ophthalmological Calendar for the Therapeutic Follow-Up of Glaucoma in the Elderly"

_ijerph, 2023, doi:10.3390/ijerph20021237_

Round 1
Reviewer 1 Report
This study aimed to build an ophthalmological calendar for the therapeutic follow-up of glaucoma in the elderly. The article introduced the establishment process of ophthalmology calendar in detail. However, this article has the following major problems:
1. The most important indicators such as intraocular pressure and visual field during the follow-up of glaucoma patients were not designed into the calendar, and it was obviously not enough to include only the information related to the use of eye drops in the calendar. Such a calendar lacks specificity for follow-up treatment of glaucoma patients.
2. Compared with previous studies, what is the difference of this study, and what are the innovations and advantages? It is not clearly stated in the article.
3. The article is not clear about the clinical significance of the ophthalmic calendar.
Author Response
|
We appreciate the careful reading of the article and the suggestions given for its improvement.
1. You're right, but the calendar was built to help the elderly in complying with the therapeutic regimen as prescribed and allow for better supervision by health professionals. This information can be found in the last paragraph of the introduction and we reinforce it in the page 4 and 6.
2. After an extensive search of the literature, no studies similar to ours were identified, so the proposed calendar is innovative and was built to respond to the needs identified in a group of elderly with glaucoma enrolled in a family health unit, as if you can see on page 2. 3. To clarify this point, we have introduced a paragraph at the beginning of the results section, page 4, as can be seen. Please see the attachment.
|

Reviewer 2 Report
Page 2 - "medication. , filling in"
Page 3 - improve legibility of Figure 1
Page 5 - improve legibility of Figure 2 (a, b)
Author Response
|
We appreciate the comments. They have been corrected. Regarding the figures, we send them in attachment, to facilitate the edition and, consequently, provide a good visualization. Please see the attachment. Regarding the two figures, it is not possible to submit with the main document. Thus, we will send them to the Editor. |

Reviewer 3 Report
This manuscript aimed to build an ophthalmological calendar for the therapeutic follow-up of glaucoma in the elderly. The authors wish to develop a dynamic consultation, and a better bond between ophthalmologists and patients. This educational material is interesting and can help elder person to trace their glaucoma treatment. There are several questions that are not addressed clearly in the manuscript.
1. Step 5: How the examining board was formed? Does the board include both ophthalmologists and elderly patients? If so, how many people in each group?
2. Where is the text for Figure A2?
3. Both figures are in a low resolution that will affect reading experience after publication.
Author Response
We appreciate the comments, thank you (research design and methods). Some changes were introduced at this levels.
The examining board is formed by two ophthalmologists and two nurses, This point was corrected.
Please see the attachment. Regarding the two figures, it is not possible to submit with the main document. Thus, we will send them to the Editor.
